# The Role of Bone-Anchored Hearing Devices and Remote Microphones in Children with Congenital Unilateral Hearing Loss

**DOI:** 10.3390/brainsci13101379

**Published:** 2023-09-28

**Authors:** Francesco Lazzerini, Luca Bruschini, Giacomo Fiacchini, Pietro Canzi, Stefano Berrettini, Francesca Forli

**Affiliations:** 1Otolaryngology, Audiology and Phoniatrics Unit, University of Pisa, 56126 Pisa, Italy; luca.bruschini@unipi.it (L.B.); giacomo.fiacchini@unipi.it (G.F.); s.berrettini@med.unipi.it (S.B.);; 2Department of Clinical and Experimental Medicine, University of Pisa, 56126 Pisa, Italy; 3Department of Surgical, Medical and Molecular Pathology and Critical Care Medicine, University of Pisa, 56126 Pisa, Italy; 4Department of Clinical, Surgical, Diagnostic and Paediatric Sciences, University of Pavia, 27100 Pavia, Italy; pietro.canzi@unipv.it; 5Clinical Science, Intervention and Technology, Karolinska Institute, 17177 Stockholm, Sweden

**Keywords:** bone conduction device, remote microphone, unilateral hearing loss, children, SSD, aural atresia

## Abstract

Congenital unilateral hearing loss (UHL) represents a contemporary audiologic challenge. Children with UHL can struggle with understanding speech in noise, localizing sounds, developing language, and maintaining academic performance, leading to low self-esteem, anxiety, and decreased social support. Two specific conditions related to UHL in children are single-sided deafness (SSD) and unilateral auris atresia (UAA). This was a retrospective observational study on a group of children with UHL. The Simplified Italian Matrix Sentence Test was used for the assessment of speech reception threshold (SRT) in different conditions: speech and noise from the front (S0N0), speech at 45° from the side of the better ear and noise at 45° from the opposite side (SbNw), and vice versa (SwNb). Each test was conducted unaided, with a bone-anchored hearing device (BAHD), and with a remote microphone (RM) system. The use of a BAHD and RM led to an improvement in SRT in S0N0 and SwNb conditions. The SSD subgroup demonstrated significant benefits with both devices in SwNb, and the UAA subgroup from the use of BAHD in S0N0. In conclusion, the study underscores the potential benefits of both devices in enhancing speech perception for UHL children, providing insights into effective intervention strategies for these challenging cases.

## 1. Introduction

Unilateral hearing loss (UHL) in children represents one of the currently most difficult audiological challenges. Firstly, this is a relatively common condition: the incidence of congenital sensorineural hearing loss is around 1–3 per 1000 births, and 30–40% of these are reported to be cases of UHL; moreover, the incidence increases with age [1,2]. Furthermore, an increasing number of studies have shown how such a condition can result in school-age children experiencing various auditory, educational, and psychosocial challenges [3,4,5,6,7,8,9].

In particular, compared to peers without hearing loss, children with UHL struggle more with understanding speech in noise, localizing sounds, developing language and cognitive skills, and maintaining academic performance [3,6,7,9,10,11,12,13,14]. UHL can also lead to low self-esteem, anxiety, strained peer relations, and decreased social support [4].

The challenges faced by individuals with UHL predominantly stem from the deficiency in spatial hearing. Spatial hearing allows the auditory system to understand the unique pathways through which sounds journey from their origins to each ear, contributing to a comprehensive perception of the auditory environment. It obviously requires two well-functioning ears. By enabling sound localization, discrimination in noisy settings, and orientation towards sound origins, spatial hearing plays a pivotal role in enhancing speech recognition in noisy or acoustically difficult environments [15].

This introduction underscores the importance of early identification and appropriate interventions to support children with UHL and help them overcome the unique obstacles they may encounter [6].

Two specific conditions related to UHL in children are single-sided deafness (SSD) and unilateral auris atresia (UAA).

SSD refers to a condition where an individual experiences significant sensorineural hearing loss in one ear while preserving normal hearing in the other ear, without the evidence of malformation of the middle or external ear structures.

UAA is a congenital condition characterized by the absence or severe underdevelopment of the external ear canal and middle ear structures in one ear. This results in a significant and usually pure conductive hearing loss on the affected side, while the other ear remains unaffected.

Different approaches have been proposed for the hearing rehabilitation of children with these specific UHL conditions. 

For pediatric patients, congenital SSD cochlear implantation (CI) represents a promising solution, especially in etiologies with a fragility of the contralateral ear (e.g., congenital CMV [16,17] or enlarged vestibular aqueduct [18,19]), and it is supposed to be the only intervention that can potentially restore binaurality. However, CI is not always feasible (e.g., for cochlear nerve aplasia or sever hypoplasia, which accounts for nearly 20% of congenital SSD [2,20]) and the age at implantation remains a controversial topic, representing a factor that can strongly limit the results [20,21,22]. 

Other valuable proposed solutions are the use of bone-anchored hearing devices (BAHD) on the deaf side ear, acting as bone conduction contralateral routing of sounds (CROS), or the use of remote microphones systems (RM) on the normal ear, especially in scholastic settings, that can guarantee an improvement in the sound-to-noise ratio (SNR), limiting the hearing impairment related to UHL. None of those solutions can restore any sort of spatial hearing in SSD, but some benefits have been widely reported in the literature [23,24,25].

RM are wireless technologies employed as assistive devices for children with hearing difficulties, designed to mitigate the adverse impacts of background noise on speech comprehension in challenging auditory settings. Previous studies have demonstrated the manifold advantages of RM. These advantages encompass enhanced speech comprehension in noisy or distant environments [26], elevated academic performance, bolstered progress in speech and language development, improved classroom behavior and attentiveness [27]. The benefits extend beyond mere speech comprehension and communication skills, positively influencing the overall well-being of the child. This includes heightened attention levels and reduced listening-effort, resulting in diminished listening fatigue [5,9,28]. In Western countries, the use of RMs is integrated into the official protocols for hearing loss rehabilitation. They are recommended for all school-aged children with hearing difficulties, both in educational and home settings, starting from early childhood [29,30].

For UAA, instead, surgical repair may not always result in a significant improvement in hearing [31,32], and revision surgeries are commonly needed [33,34,35]. Therefore, BAHD (on soft-band, or adhesive adapter [36], as well as with percutaneous or transcutaneous abutment), the Vibrant Bonebridge (MED-EL GmbH) [37,38], the Vibrant Soundbridge (MED-EL GmbH, Innsbruck, Austria) [39], and the Osia II (Cochlear Corp., Sydney, Australia) [40,41] have also been offered to children with UAA, with the aim of providing binaural hearing [42,43]. Even in UAA cases, RM can provide hearing relief, especially in difficult situations such as a scholastic setting.

The aim of the present study is to investigate the benefits in terms of speech perception in noise, with the use of BAHD and RM, in a cohort of children with congenital UHL. Moreover, this cohort presents two subgroups: the first subgroup includes SSD children, while the second group is formed by children with UAA. Differences in benefits within the subgroups will also be analyzed. 

The experimental conditions will include situations with speech and noise originating from the same loudspeaker, and situations with speech and noise originating from different sound sources.

The results derived from this study could enhance the existing knowledge on the listening in noise features in children with congenital UHL and the effects of BAHD and RM on this ability.

## 2. Materials and Methods

This is a retrospective observational study on a group of patients with UHL followed by an audiology service in a third-level referral University hospital.

Patients were voluntarily enrolled in the study during the routine visits for audiologic assessment in the pediatric audiology clinic. 

Enrolment criteria were the following:

Age < 16 years and >5 years;No previous experience with hearing devices or RM;Appropriate development of speech and language;Presence of SSD (defined as unilateral sensorineural hearing loss with a pure tone average hearing threshold at 250 Hz, 500 Hz, 1000 Hz, and 2000 Hz (PTA) ≥ 65 Db, with a contralateral PTA < 30 dB) or presence of UAA.

Exclusion criteria were the following:

Multihandicap and/or cognitive impairment.

The audiologic assessment was conducted by clinicians and technicians experienced in the management of pediatric hearing loss. Audiological evaluation included the collection of the medical history, otoscopy, pure-tone audiometry, speech recognition threshold (SRT), and a squelching abilities assessment. 

All tests were conducted in a 5.2 m × 3.5 m sound-isolated room, using an Otometrics MADSEN Astera2 audiometer with a set of TDH39 headphones and Indiana Line loudspeakers.

The SRT was tested with the Simplified Italian Matrix Sentence Test (SIMST), which is an adaptive speech audiometry in background noise with an automatically variated signal to noise ratio (SNR). It is specific for children aged 4 years or more and uses 14 randomized lists of three-word sentences with the same structure (number, adjective, noun). The background noise is presented at a fixed 65 dB sound pressure level (SPL), while the speech material is actively adapted via the software. The result of the test is a score representing the SRT: the SNR in dB at which the patient can recognize 50% of the speech material [44].

Following the evaluation paradigm used in previous studies by our and other groups [45,46], the SIMST was utilized to assess SRT and squelch ability, both with and without hearing devices (BAHD and RM). The starting speech level was set at 65 dB, and the fixed noise level was set at 65 dB. Stimuli were presented in three different conditions: ‘S0N0’ (speech and noise from 1 m in front), ‘S +45° N −45°’ (speech from 1 m at 45° to the right and noise from 1 m at 45° to the left of the patient’s head), and ‘S −45° N +45°’ (speech from 1 m at 45° to the left and noise from 1 m at 45° to the right of the patient’s head). We labeled these conditions as ‘S0N0’ when both the speech and noise were presented from a single speaker in front of the patient, ‘SbNw’ when the speech was presented from the better side and the noise from the worse side, and ‘SwNb’ when the speech was presented from the worse side and the noise from the better side. The experimental setting is schematized in Figure 1.

All the patients had previously been tested in all the 3 conditions without hearing devices, then with BAHD on a soft band on the deaf side, finally with a RM hung 15 cm below the loudspeaker emitting the speech signal and connected to a non-amplified hearing aid with an in-the-canal (ITC) receiver, applied on the normal ear. A schematic representation of this setting is presented in Figure 2. Before the evaluation, all the children underwent 2 or 3 practice trials with the Matrix Sentence Test, to make them comfortable with this exam.

The tested BAHD was a Ponto 5 Superpower (Oticon Medical Corp., Askim, Sweden), placed on a soft band and fitted according to the hearing level of each patient. The tested RM system was composed of a GN Resound Corp., Ballerup, Denmark. MicroMic, remotely connected to a GN Resound Corp. Key RIE 61 retroauricular hearing aid with receiver in the ear.

This study was approved by the Institutional Ethics Committee. The research was conducted ethically, with all study procedures performed in accordance with the requirements of the World Medical Association’s Declaration of Helsinki. The parents of the children all gave written informed consent for study participation and data publication.

Moreover, the application of BAHD and the use of RM is a regular health care treatment option in Italy and all testing data were part of regular follow-up measurements, which were used to optimize the hearing outcomes.

### Statistical Analysis

A comprehensive data analysis was conducted using the statistical software IBM SPSS Statistics 26 (IBM Corp., Armonk, NY, USA). For group comparisons, the non-parametric Mann–Whitney U test and the Wilcoxon signed rank test were utilized given the small number of the study population. When assessing relationships between dichotomous variables (variables with two categories or levels) we used the Chi-square test. The significance level for all tests was set at *p* < 0.05. For the univariate analysis, the correlations between continuous, categorical, and dichotomous variables were evaluated using the appropriate techniques, including the Pearson correlation coefficient for linear relationships and the Spearman rank correlation coefficient for non-linear relationships. Again, all correlations were tested for statistical significance at *p* < 0.05. 

## 3. Results

Ten children with SSD or UAA were enrolled in this study. 

The Demographic data and audiologic characteristics of the cohort are reported in Table 1.

In all cases the hearing loss was diagnosed at birth via newborn hearing screening (the detailed protocol of hearing screening in our department has been previously published by our group [16,47]). The otoscopy was bilaterally normal in all the children of the SSD subgroup, while in the UAA subgroup it was normal on the non-atretic side. however, it was possible to assess various grades of aural atresia on the opposite side.

SRT results in all the configurations in the whole sample, as well as in both the subgroups, are reported in Table 2. Squelching abilities refer to SRT in the SbNw and SwNb conditions. 

### 3.1. UHL Population

Analyzing the whole sample in the unaided configuration, the SRT in SbNw (−6.6 dB HL [−9.4 ÷ −2.9]) was statistically better than in SwNb (−0.9 dB HL [−3.7 ÷ 2.7]) (*p* = 0.005). 

#### 3.1.1. BAHD vs. Unaided

With the use of the BAHD, it was possible to notice a significant improvement in SRT in the S0N0 configuration (−3.4 dB HL [−6.1 ÷ 2.3]) in comparison to when unaided (−1.9 dB HL [−4.3 ÷ 1.7]) (*p* = 0.037). It was also possible to assess a slightly significant worsening of SRT in the SbNw configuration with the BAHD (−5.1 dB HL [−8.4÷ 2.4]) compared to when unaided (−6.6 dB HL [−9.4 ÷ −2.9]) (*p* = 0.047), but there was a significant improvement in SRT in the SwNb configuration (−3.7 dB HL [−7 ÷ 0.1]) in comparison to when unaided (−0.9 dB HL [−3.7 ÷ 2.7]) (*p* = 0.009). 

It was remarkable to notice that with the BAHD the mean SRT in SbNw (−5.1 dB HL [−8.4 ÷ 2.4]) and SwNb (−3.7 dB HL [−7 ÷ 0.1]) lost their statistically significant difference (*p* = 0.057). 

#### 3.1.2. RM vs. Unaided

Even with the use of the RM, the SRT in S0N0 (−3.3 dB HL [−7.5 ÷ −0.2]) result was significantly improved, in comparison to when unaided (−1.9 dB HL [−4.3 ÷ 1.7]) (*p* = 0.028). The slight improvement of the SRT in SbNw with the RM (−7.5 dB HL [−12.1 ÷ −1.3]), vs. when unaided (−6.6 dB HL [−9.4 ÷ −2.9]) was not significant (*p* = 0.202), while the improvement with the RM was statistically significant in SwNb (−5.5 dB HL [−12.3 ÷ −1]) when compared with the unaided condition (−0.9 dB HL [−3.7 ÷ 2.7]) (*p* = 0.007).

#### 3.1.3. BAHD vs. RM

No statistically significant differences were found between the mean SRT with the BAHD and with the RM in the S0N0 configuration (−3.4 dB HL [−6.1 ÷ 2.3] and −3.3 dB HL [−7.5 ÷ −0.2], respectively) (*p* = 0.760), in SbNw (−5.1 dB HL [−8.4 ÷ 2.4] and −7.5 dB HL [−12.1 ÷ −1.3], respectively) (*p* = 0.074), or in SwNb (−3.7 dB HL [−7 ÷ 0.1] and −5.5 dB HL [−12.3 ÷ −1], respectively) (*p* = 0.285).

Unlike the use of the BAHD, with the RM the mean SRT in SbNw (−7.5 dB HL [−12.1 ÷ −1.3]) remained significantly better than in the SwNb configuration (−5.5 dB HL [−12.3 ÷ −1]) (*p* = 0.028).

### 3.2. SSD Subgroup

In the SSD subgroup, the mean unaided SRT in SbNw (−5.9 dB HL [−8.5 ÷ −2.9]) was found to be significantly better than in SwNb (0.9 dB HL [−0.9 to 2.7]) (*p* = 0.043).

#### 3.2.1. BAHD vs. Unaided

In the SSD subgroup, the utilization of the BAHD demonstrated a minor improvement that lacked statistical significance in the S0N0 configuration (−2.3 dB HL [−5.4 ÷ 2.3]) compared to the unaided condition (−0.9 dB HL [−1.9 ÷ 1.7]) (*p* = 0.225). On the other hand, the use of the BAHD revealed a slightly significant deterioration in SRT in the SbNw configuration (−3.4 dB HL [7.9 ÷ 2.4]) as opposed to the unaided condition (−5.9 dB HL [−8.5 ÷ −2.9]) (*p* =.042). However, a noteworthy improvement in SRT within the SwNb condition (−2.8 dB HL [−4.8 ÷ −0.1]) was observed with the BAHD compared to the unaided condition (0.9 dB HL [−0.9 ÷ 2.7]) (*p* = 0.042).

In this subgroup, additionally, the difference between SRT in the SbNw (−3.4 dB HL [−7.9 ÷ 2.4]) and SwNb (−2.8 dB HL [−4.8 ÷ −0.1]) configurations did not become statistically significant when using the BAHD (*p* = 0.345). 

#### 3.2.2. RM vs. Unaided

In children with SSD, the use of the RM in the S0N0 configuration achieved an improvement in SRT (−2.0 dB HL [−3.8 ÷ −0.2]) with respect to the unaided condition (−0.9 dB HL [−1.9 ÷ 1.7]), even if not statistically significant (*p* = 0.223). Furthermore, it was also noticeable that in the SbNw configuration, the use of the RM achieved an improvement in SRT (−6.3 dB HL [−9.9 ÷ −1.3]) rather than in the unaided condition (−5.9 dB HL [−8.5 ÷ −2.9]), but did not reach statistical significance (*p* = 0.500). In the more difficult SwNb configuration, SRT with the RM improved (−4.2 dB HL [−8 ÷ −1]) in comparison to the unaided condition (0.9 dB HL [−0.9 ÷ 2.7]), reaching a statistically significant difference (*p* = 0.043).

#### 3.2.3. BAHD vs. RM

In the SSD subgroup too, no significant differences were found in SRT with the BAHD and with the RM in the S0N0 configuration (−2.3 dB HL [−5.4 ÷ 2.3] and −2.0 dB HL [−3.8 ÷ −0.2], respectively) (*p* = 0.500), in the SbNw configuration (−3.4 dB HL [−7.9 ÷ 2.4] and −6.3 dB HL [−9.9 ÷ −1.3], respectively) (*p* = 0.225), or in the SwNb configuration (−2.8 dB HL [−4.8 ÷ −0.1] and −4.2 dB HL [−8 ÷ −1], respectively) (*p* = 0.500).

### 3.3. UAA Subgroup

Even in the UAA subgroup, the mean unaided SRT in the SbNw configuration (−7.3 dB HL [9.4 ÷ −4.8]) was found to be significantly better than in the SwNb configuration (−2.7 dB HL [−3.7 ÷ −1.2]) (*p* = 0.043). 

#### 3.3.1. BAHD vs. Unaided

Conversely to the SSD subgroup, in UAA, the use of the BAHD led to a significant improvement in SRT in the S0N0 configuration (−4.7 dB HL [−6.1 to −3.2]) except when unaided (−2.8 dB HL [−4.3 ÷ 0]) (*p* = 0.042). The mean SRT in the SbNw configuration with the BADH (−6.9 dB HL [−8.4 ÷ −5.1]) was not significantly worse compared to when unaided (−7.3 dB HL [−9.4 to −4.8]) (*p* = 0.500). Furthermore, even if it was possible to describe an improvement of SRT in the SwNb configuration with the BAHD (−4.6 dB HL [−7.0 ÷ −1.3]) with respect to when unaided (−2.7 dB HL [−3.7 ÷ −1.2]), it lacked statistical significance (*p* = 0.136).

In this subgroup, the difference in the mean SRT in the SbNw (−6.9 dB HL [−8.4 ÷ −5.1]) and SwNb (−4.6 dB HL [−7.0 ÷ −1.3]) configurations remained statistically significant using the BAHD (*p* = 0.043). 

#### 3.3.2. RM vs. Unaided

In the S0N0 configuration, the SRT with the RM (−4.7 dB HL [−7.5 ÷ −2]) was better than in the unaided condition (−2.8 dB HL [−4.3 ÷ 0]), even if it was not statistically significant (*p* = 0.080). In the SbNw configuration, the SRT with the RM was slightly worsened (−6.9 dB HL [−8.4 ÷ −5.1]) compared to the unaided condition (−7.3 dB HL [−9.4 ÷ −4.8]), but again the difference was not statistically significant (*p* = 0.225). On the other hand, in the more challenging SwNb configuration, the SRT with the RM (−6.8 dB HL [−12.3 ÷ −3.4]) was found to be increased in comparison to the unaided condition (−2.7 dB HL [−3.7 ÷ −1.2]), but with no statistically significant difference (*p* = 0.080). 

#### 3.3.3. BAHD vs. RM

No significant differences were found in SRT with the BAHD and with the RM in the S0N0 configuration (−4.7 dB HL [−6.1 ÷ −3.2] and −4.7 dB HL [−7.5 ÷ −2], respectively) (*p* = 0.893), in the SbNw situation (−6.9 dB HL [−8.4 ÷ −5.1] and −8.7 dB HL [−12.1 ÷ −4.7], respectively) (*p* = 0.225), or in the SwNb configuration (−4.6 dB HL [−7.0 ÷ −1.3] and −6.8 dB HL [−12.3 ÷ −3.4], respectively) (*p* = 0.345).

A graphic representation of the significant results is reported in Figure 3, Figure 4 and Figure 5.

Age did not significantly correlate with SRT outcomes, nor did the mean PTA of AC or BC in the better or worse ear.

## 4. Discussion

The results of the study shed light on the impact of two distinct hearing treatment aids, the BAHD and RM, on SRT in different configurations for children with UHL.

It is well documented that children with UHL can experience significant hearing difficulties. In particular, hearing asymmetry can contribute to the impairment of spatial hearing, leading to challenges in comprehending speech amidst noise, squelching ability, localizing sounds accurately, acquiring language and cognitive abilities, and upholding academic achievements [3,6,7,9,10,11,12,13,14]. Moreover, this condition has been connected to issues such as diminished self-esteem, heightened anxiety, strained peer interactions, and reduced social support [20,21].

Our experimental data confirmed that SRT is higher in individuals with UHL compared to healthy individuals. In fact, we were able to assess a mean SRT in the SIMST in the S0N0 condition of −1.9 dB HL, well lower than the −6.2 dB HL ± 1.3 dB HL considered the normal value in the typical pediatric hearing population [44].

Moreover, it was also possible to measure a significant impairment of squelch abilities, with rather good performances when the speech signal came from the normal hearing side and the noise from the worse hearing side (SbNw), but very poor results when the speech material came from the worse side and the noise from the better side (SwNb). This has already been reported in the scientific literature, and also in different categories of hearing-impaired subjects [45,46]. In our cohort, in fact, when unaided, there was a strong and statistically significant difference between the mean SRT in the SbNw configuration and the mean SRT in the SwNb configuration.

The same evidence could be found in both the tested subgroups. Despite this, children with UAA presented significantly slightly better results in the more demanding SwNb configuration, compared to the SSD population. This could be due to the significantly better residual hearing in the worse ear.

The use of a BAHD could lead to improved speech understanding in noise in the pediatric UHL population, at least in certain listening conditions. In fact, the test of the prosthetic device on the worse hearing side on a soft band led to a significantly better SRT in the S0N0 configuration, but mostly to a significant improvement of the SRT in the SwNb configuration. In analyzing the subgroups, this effect was found to be significant only in the SwNb configuration for the SSD subgroup, while only in the S0N0 configuration in the UAA population.

These findings are partially in line with previous research, both for the SSD [23,48] and UAA subgroups [49,50].

Concerns about the use of the BAHD as a contralateral route of signals (CROS) in the SSD population have been reported, especially for the abolition of the head shadow effect [24,25]. In particular, the loss of the head shadow effect associated with the use of CROS in subjects with UHL can be expected to cause a worsening in speech understanding in situations when the speech signal comes from the better hearing side and the noise from the contralateral side (a situation that we experimentally reproduced as SbNw). In the cohort that we analyzed, a slight worsening of the SRT in the SbNw configuration occurred with the use of the BAHD, and it was significant in the SSD subgroup.

Of course, it is clear and widely accepted that the use of a BAHD in SSD could not give rise to binaurality [23,51,52]. On the other hand, more intriguingly, it could be speculated that the use of a BAHD in the UAA cohort could lead to a restored binaurality, given the expected AC/BC gap closure [53,54,55]. From our results it seems to be arguable, at least in our study population with UAA. In fact, even if the results in the SbNw configuration with the BAHD were not significantly lower than unaided, the lack of an improved SRT in this type of setting can exclude the presence of some sort of binaural unmasking. This aspect has previously been discussed by some authors [31]. The reasons for this lack of spatial hearing recovery could be related to many factors, mainly because BAHD offers auditory information that is less valuable and precise in comparison to the normal hearing ear on the opposite side and the conduction of sound through bone is facilitated by the skull bone itself, which can lead to interactions between the air–bone signal and cross-stimulation from the BAHD in the normal hearing cochlea [56]. However, the lack of an early stimulation of the atretic ear with the BAHD, as well as insufficient training with the device could also limit the access to binaural abilities. In fact, it is highly likely that individuals with unilateral atresia could be compelled towards unilateral hemispheric dominance if the ear with atresia does not receive sufficient stimulation during the early years of life [31,57]. This could potentially result in a diminished perceived advantage when utilizing a hearing aid.

On the other hand, a recent article by Vogt et al. [58] indicated that undergoing implantation between the ages of 4 to 6 years does not yield superior performance outcomes when compared to the 6 to 10 year age range for implantation. It is important to note that these findings are limited in their applicability due to their reliance on a relatively small dataset. Consequently, further studies involving children with conductive UHL who receive amplification through a BAHD before the age of 4 should be carried out.

In our study the use of the RM was associated with a significant improvement in SRT both in the S0N0 and SwNb configurations. Analyzing the subgroups independently, the significant improvement was limited to the SwNb configuration in children with SSD. In other cases, even if it was possible to assess improvements, the differences in the unaided condition were not statistically significant in the S0N0 configuration in the SSD subgroup, or in the S0N0 and SbNw configurations in the UAA subgroup. On the other hand, unlike the BAHD, the use of the RM did not cause a significant worsening of the results.

The use of a RM has been reported to be associated with improvements in various hearing-impaired populations [30,59,60]. Moreover, RM systems are reported to have the advantage of not eliminating the head shadow effect, avoiding the significant reduction in speech perception when the signal comes from the better hearing side and the noise contralaterally (SbNw), which is typical of CROS systems [60]. Our study confirmed that the RM is a valuable device in limiting the impairment derived from a congenital UHL.

No significant differences were found between the use of a BAHD or RM in terms of SRT improvement in various settings of stimuli administration. However, the better SRT results in S0N0, SbNw, and SwNb were obtained with the RM (see Figure 5).

From the results of our study, it was possible to underline the usefulness of the BAHD and RM in the hearing amelioration of a population of UHL children.

This population and its subgroups of SSD and UAA may represent a challenging group of patients to treat, especially when the candidacy to CI is arguable (e.g., due to a high age at evaluation, as well as parental refusal of the procedure), or when the atretic ear may not lead to good hearing outcomes.

Given the contemporary difficulties associated with the presence of UHL, clinicians may offer some evidence-based solutions to relieve the problem.

As we have shown, both the BAHD and RM devices led to an improvement in SRT in different testing conditions, both in children with SSD and UAA. Even if it was not possible to define a statistically significantly better solution, each device presents pros and cons.

The BAHD can be worn on a soft band as well as being fitted through a minor surgical procedure. In our cohort, its use showed a significant benefit in terms of SRT in the S0N0 (also in the SSD and UAA subgroups) and SwNb configurations (also in the SSD subgroup). On the other hand, the elimination of the head shadow effect due to the contralateral routing of signal resulted in a slight reduction in SRT performance in the SbNw configuration. Even if encouraging, the results with such a group of protheses remain widely variable [31,35,49,50,61,62], and the study conducted by Nelissen et al. found that a substantial percentage of children with congenital unilateral conductive hearing loss, around 50%, discontinued the use of their BAHD within a few years (less than 5 years) of implantation [63]. Moreover, given the aesthetic and emotional impact correlated with the use of prosthetic devices, not all parents accept the adoption of such hearing rehabilitation options. In those cases, the use of a RM can be an effective way of limiting the difficulties of hearing in noisy environments for children with UAA, as well as for children with SSD.

Indeed, RMs represent a very easy-to-use aid, since they do not require surgery and can be used just when more hearing support is needed (e.g., in a school environment). As far as the BAHD as concerned, it considered a significant benefit in terms of SRT in the S0N0 and SwNb configurations. The use of the RM in our test was further associated with the best overall results in all the tested situations in both subgroups, even if it did not yield a statistically significant amelioration with respect to the unaided or the BAHD condition. In contrast to the use of the BAHD, it was possible to prevent a decrease in SRT performance in the SbNw configuration. On the other hand, hearing scenarios are often dynamic, and such technology can limit any hearing improvement when there are many speakers present [60,64].

Finally, current BAHDs offer enhanced connectivity options through wireless technology. This feature holds significant potential, especially in the pediatric population, as it enables the linkage of BAHDs and RM systems. So, in educational settings, educators can effectively utilize these capabilities to facilitate interactions with multiple devices and enhance the learning environment. The aforementioned technology has the potential to enable the pediatric population with UHL to harness the advantageous effects of both a BAHD and RM together.

### Study Limitations

The study has several limitations that should be considered when interpreting the results.

First, the sample size was small, which could affect the generalizability of the findings to a larger population.

Additionally, the retrospective design of the study might introduce bias and affect the accuracy of the data collection and analysis. Furthermore, the study primarily focused on the short-term outcomes of using a BAHD and RM, and did not analyze the potential long-term effects. Moreover, the study was conducted in a controlled laboratory setting, which may not fully reflect real-world conditions. The findings should be cautiously extrapolated to everyday environments, which often involve complex and dynamic situations. Finally, the absence of a control group of normal hearing subjects must be considered.

Despite these limitations, the study provides interesting insights into the determination of the effect of some devices on speech perception in noise in specific populations of children with UHL, but further research addressing these limitations is necessary to confirm and extend our findings.

## 5. Conclusions

The findings of this study emphasize the importance of tailored interventions for children with UHL to address their unique auditory needs. Both the BAHD and RM hold promise as valuable tools in improving speech perception, providing potential solutions for the challenges posed by UHL. Our findings indicate that it would be important to contemplate early rehabilitation with the BAHD and/or RM in the UHL pediatric population and discuss this matter with the patients’ families immediately after the diagnosis. Further research and clinical trials are warranted to continue refining these interventions and optimizing the outcomes for this population.

## Figures and Tables

**Figure 1 brainsci-13-01379-f001:**
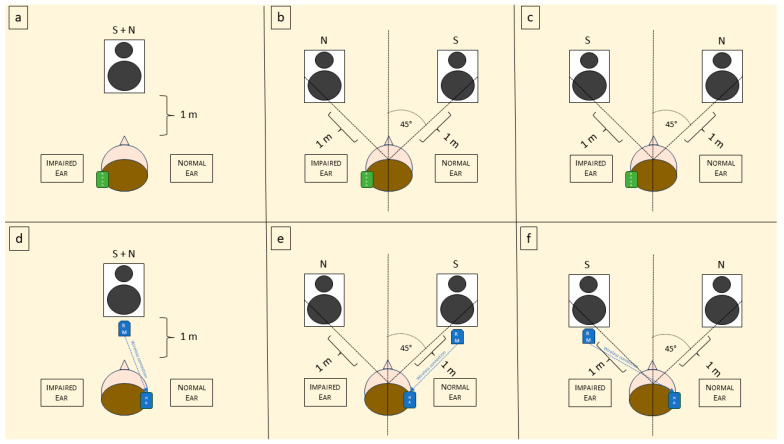
Schematic representation of the experimental setting. S = sound; N = noise; BAHD = bone-anchored hearing device; RM = remote microphone; HA = hearing aid. (**a**) Representation of the use of BAHD in S0N0 condition; (**b**) use of the BAHD in SbNw; (**c**) use of the BAHD in SwNb; (**d**) representation of the use of RM in S0NO condition; (**e**) use of RM in SbNw; and (**f**) use of RM in SwNb.

**Figure 2 brainsci-13-01379-f002:**
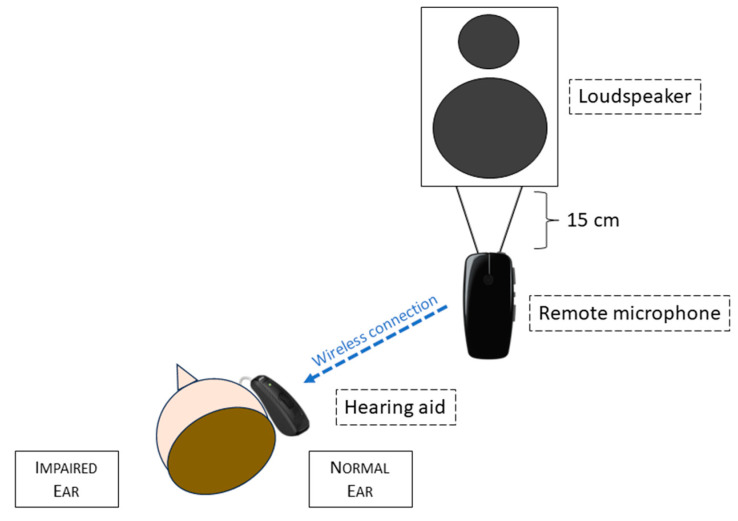
Diagram of the remote microphone system operating in our experimental setting.

**Figure 3 brainsci-13-01379-f003:**
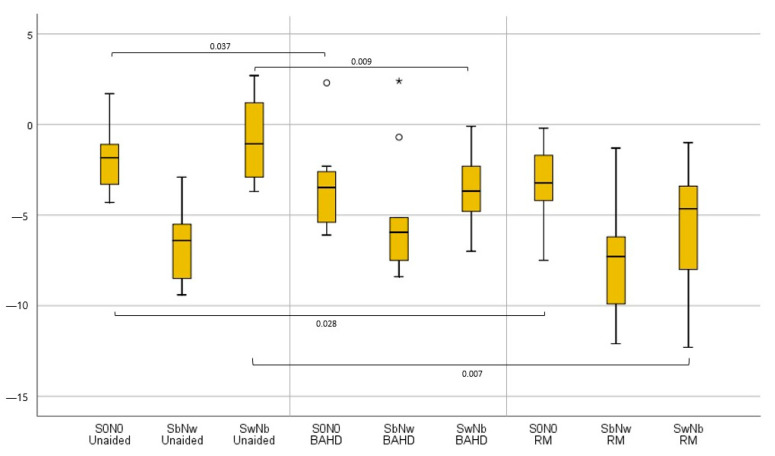
Boxplot graphic representation of tested UHL children’s results in terms of SRT in the different testing conditions; when unaided, with the BAHD, and with the RM. Outliers were reported with the ° marks, and very outliers with * marks.

**Figure 4 brainsci-13-01379-f004:**
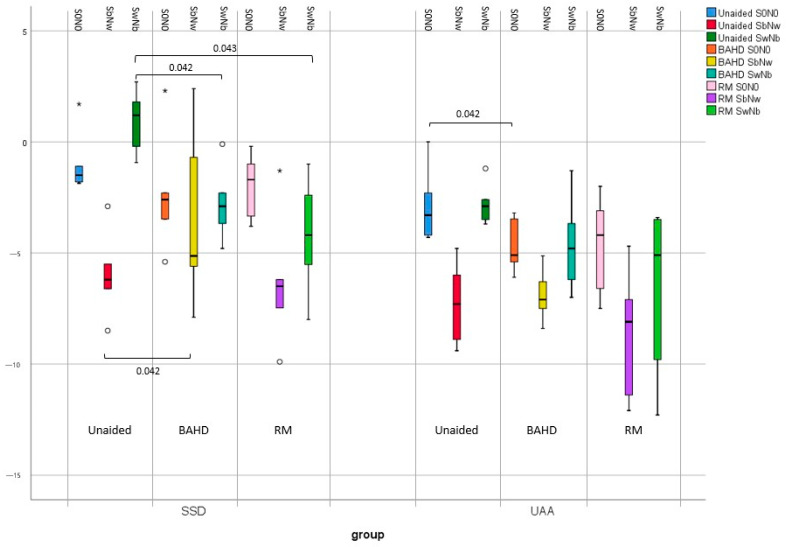
Boxplot graphic representation of results in terms of SRT in the different testing conditions; when unaided, with the BAHD, and with the RM, in the SSD and UAA subgroups. Outliers were reported with the ° marks, and very outliers with * marks.

**Figure 5 brainsci-13-01379-f005:**
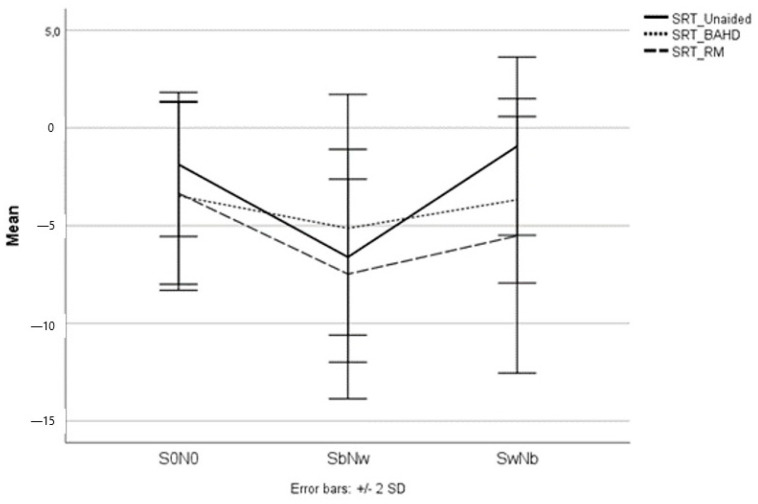
The mean SRT scores of the pediatric UHL study population in all the testing conditions; when unaided, with the BAHD, and with the RM.

**Table 1 brainsci-13-01379-t001:** Demographic and audiologic characteristics of the cohort and the subgroups.

	Total	SSD	UAA	*p*
Mean Age (y)	10.2 (5 ÷ 15)	9.6 (5 ÷ 15)	10.8 (8 ÷ 13)	0.575
Gender (F; M)	3; 7	2; 3	1; 4	0.500
Hearing-impaired side (R; L)	2; 8	2; 3	0; 5	0.444
Hearing threshold better ear (AC) (dB HL)	17.2 (15 ÷ 20)	18.7 (15 ÷ 20)	16 (15 ÷ 20)	0.125
Hearing threshold worse ear (AC) (dB HL)	62.7 (55 ÷ 85)	80 (65 ÷ 105)	58 (55 ÷ 65)	0.017 *
Hearing threshold better ear (BC) (dB HL)	17.2 (15 ÷ 20)	18.7 (15 ÷ 20)	16 (15 ÷ 20)	0.125
Hearing threshold worse ear (BC) (dB HL)	N.T.	N.T.	15 (15÷15)	

Note: the * mark underscore the statistical significance.

**Table 2 brainsci-13-01379-t002:** SRT data in S0N0, SbNw, and SwNb of the cohort and the subgroups.

	Total	SSD	UAA	*p*
Unaided				
Unaided SRT S0N0 (dB HL)	−1.9 [−4.3 ÷ 1.7]	−0.9 [−1.9 ÷ 1.7]	−2.8 [−4.3 ÷ 0]	0.095
Unaided SRT SbNw (dB HL)	−6.6 [−9.4 ÷ −2.9]	−5.9 [−8.5 ÷ −2.9]	−7.3 [−9.4 ÷ −4.8]	0.421
Unaided SRT SwNb (dB HL)	−0.9 [−3.7 ÷ 2.7]	0.9 [−0.9 ÷ 2.7]	−2.7 [−3.7 ÷ −1.2]	0.008 *
BAHD				
SRT S0N0 with BAHD (dB HL)	−3.4 [−6.1 ÷ 2.3]	−2.3 [−5.4 ÷ 2.3]	−4.7 [−6.1 ÷ −3.2]	0.151
SRT SbNw with BAHD (dB HL)	−5.1 [−8.4 ÷ 2.4]	−3.4 [−7.9 ÷ 2.4]	−6.9 [−8.4 ÷ −5.1]	0.151
SRT SbNw with BAHD (dB HL)	−3.7 [−7 ÷ 0.1]	−2.8 [−4.8 ÷ −0.1]	−4.6 [−7.0 ÷ −1.3]	0.222
RM				
SRT S0N0 with RM (dB HL)	−3.3 [−7.5 ÷ −0.2]	−2.0 [−3.8 ÷ −0.2]	−4.7 [−7.5 ÷ −2]	0.095
SRT SbNw with RM (dB HL)	−7.5 [−12.1 ÷ −1.3]	−6.3 [−9.9 ÷ −1.3]	−8.7 [−12.1 ÷ −4.7]	0.310
SRT SbNw with RM (dB HL)	−5.5 [−12.3 ÷ −1]	−4.2 [−8 ÷ −1]	−6.8 [−12.3 ÷ −3.4]	0.421

Note: the * mark underscore the statistical significance.

## Data Availability

Data will be given by the authors upon reasonable request.

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
