# Peer review of "The Role of Bone-Anchored Hearing Devices and Remote Microphones in Children with Congenital Unilateral Hearing Loss"

_brainsci, 2023, doi:10.3390/brainsci13101379_

Round 1

Reviewer 1 Report

1-      While the title of the article pertains to the perception of the language of children with hearing disorders, the subsequent discussion delves into investigations involving environmental noise disorders and sound orientation in auditory perception. It appears prudent to select a more suitable title that accurately reflects the comprehensive scope of the research.

2-      To enhance the quality of the report, consider incorporating a schematic diagram or a comprehensive visual representation of the recording and playback system setup. This visual aid should encompass multiple angles of audio playback, particularly focusing on the bone-anchored hearing device (BAHD) and its integration with a remote microphone (RM) system.

3-      The introduction section should explicitly articulate the study's objectives to provide a clear and concise roadmap for readers.

4-      In line 19, could you please elaborate on the summary of SRT?

5-      Provide a comprehensive and clear explanation of the role of the remote microphone (RM) system in the context of the study.

6-      What is the summary or overview of PTA? Please clarify this in the manuscript.

7-      It is recommended to present the results pertaining to the evaluations discussed in lines 104 and 105 in the form of a well-organized table.

8-      To enhance clarity, consider using a schematic diagram or visual representation to elucidate the components of the hearing aid system as outlined in lines 129 to 133.

9-      The study's current participant count is relatively small, potentially limiting its statistical power. To enhance robustness, it is advisable to increase the number of participants.

10-  The statistical analysis report requires revision to ensure it is presented in an appropriate and comprehensible manner.

11-  Please address the resolution issue with Figure 2 to ensure its suitability for clear interpretation.

12-  In the context of this study, it would be valuable to specify which items mentioned in lines 283 to 285 were evaluated and compared.

13-  Correct lines 290 and 291 to read: "SRT is higher in individuals with unilateral hearing loss (UHL) compared to healthy individuals."

14-  Ensure that the study's objectives are explicitly and clearly stated in both the introduction and discussion sections. Furthermore, align the discussion segment of the article with the enumerated items mentioned in the manuscript.

Minor editing of English language required.

Author Response

We thank the reviewer for its kind work and for the suggestions.

We believe that according to its advice, the paper can be deeply improved and be suitable for publication.

Point-by-point responses will follow in bold. Changes in text will be in Track Change format. 

1-      While the title of the article pertains to the perception of the language of children with hearing disorders, the subsequent discussion delves into investigations involving environmental noise disorders and sound orientation in auditory perception. It appears prudent to select a more suitable title that accurately reflects the comprehensive scope of the research.

We agree with the reviewer on this point. A more accurate title has been worked out.

2-      To enhance the quality of the report, consider incorporating a schematic diagram or a comprehensive visual representation of the recording and playback system setup. This visual aid should encompass multiple angles of audio playback, particularly focusing on the bone-anchored hearing device (BAHD) and its integration with a remote microphone (RM) system.

The experimental setting has been schematized in a figure: new figure 1. All the other figures have been renumbered accordingly. See lines 152 to 157. 

3-      The introduction section should explicitly articulate the study's objectives to provide a clear and concise roadmap for readers.

The latter part of introduction presented the study aims. At lines 103 to 107 this aspect has been more thoroughly written. 

4-      In line 19, could you please elaborate on the summary of SRT?

Done.

5-      Provide a comprehensive and clear explanation of the role of the remote microphone (RM) system in the context of the study.

Done. See lines 79 to 90 and 406 to 412.

6-      What is the summary or overview of PTA? Please clarify this in the manuscript.

Done. See lines 117-118.

7-      It is recommended to present the results pertaining to the evaluations discussed in lines 104 and 105 in the form of a well-organized table.

We agree with the reviewer. Many of the data regarding the audiologic evaluation was already tabulated. In fact, table 1 and 2 contain data derived from the medical history as well as SRT and squelch abilities (reported as SRT when Speech and Noise came from different sources, on the opposite side of the listener head). 

Anyway, data regarding the age at hearing loss diagnosis, as well as otoscopy had been added at lines 203 to 207.  

8-      To enhance clarity, consider using a schematic diagram or visual representation to elucidate the components of the hearing aid system as outlined in lines 129 to 133.

Done. See figure 2 (other figures have been renumbered accordingly). 

9-      The study's current participant count is relatively small, potentially limiting its statistical power. To enhance robustness, it is advisable to increase the number of participants.

We agree with the reviewer. The cohort of the present study is quite small, even if not lot smaller than population of similar studies (see, for example, Brotto 2023).

Given the relative infrequent conditions, for the present work, it would be impossible to increase the number of study population. This aspect have been already discussed in "study limitations" section. 

10-  The statistical analysis report requires revision to ensure it is presented in an appropriate and comprehensible manner.

We double checked the statistical report. We found it flawless, even if not immediate to be understood, due the numerous data. Some grammar correction have been made to make that more comprehensible. 

11-  Please address the resolution issue with Figure 2 to ensure its suitability for clear interpretation.

Figure 2 resolution has been increased. Editors could indicate if it could be enough for achieving clarity in the printing version of the manuscript, if it will be eventually published.

12-  In the context of this study, it would be valuable to specify which items mentioned in lines 283 to 285 were evaluated and compared.

It is well discussed in the paper that that speech perception in noise and squelching abilities have been evaluated in our population. The reviewer can found it at lines 103 to 107 and 341 to 357. 

13-  Correct lines 290 and 291 to read: "SRT is higher in individuals with unilateral hearing loss (UHL) compared to healthy individuals."

Done.

14-  Ensure that the study's objectives are explicitly and clearly stated in both the introduction and discussion sections. Furthermore, align the discussion segment of the article with the enumerated items mentioned in the manuscript.

Done.

-------

We want to thank the reviewer again for its work of revision. We hope that editing the paper according to the reviewer suggestion may be suitable for publication. 

Reviewer 2 Report

Overall, this paper " Speech Perception in Children with Congenital Unilateral Hearing Loss: The Role of Bone Anchored Hearing Devices and Remote Microphones  " describes how children with a unilateral HL do with either a BAHA device or a remote microphone re. speech perception. Overall, the manuscript should be read by a native English speaking medical writer. The English is not too bad but there are grammatical mistakes.

INTRODUCTION

I want to stress that BAHAs or remote microphones do not restore binaural hearing. It is not clear to me that authors actually understand that detail. The only intervention that can potentially do that is a cochlear implantation in the affected ear.

It is also not completely clear to me the distinction between what the authors call the SSD group and the UAA group. Did CT scanning show that in the SSD group the middle and outer ear was totally normally formed. The manuscript doesn’t state anything about this.

METHODS

You don’t need to state the exclusion criteria. They are literally the opposite from your inclusion criteria.

When you use SRT for the first time you have to say there what it stands for, not a couple of sentences later….

In this context you don’t talk about dB. It doesn’t mean anything. Was it dBSPL? Or maybe dBHL if your audiometer is calibrated in that way…

Also make sure that when you talk about BAHA or remote microphone you talk about a hearing DEVICE. Talking about a hearing aid makes it very confusing.

RESULTS

The analysis is only done with 10 individuals, 5 in each group. This is just too small to allow for any parametric analysis, no matter if you checked for normal distribution or not…  

needs to be read by an Native English Speaker. Especially in the introduction, there are grammatic errors.

Author Response

We thank the reviewer for its comments, its work of revision and for the kind suggestions.

Point-by-point responses will follow in bold. Changes in text will be in Track Change format. 

R- Overall, this paper " Speech Perception in Children with Congenital Unilateral Hearing Loss: The Role of Bone Anchored Hearing Devices and Remote Microphones  " describes how children with a unilateral HL do with either a BAHA device or a remote microphone re. speech perception. Overall, the manuscript should be read by a native English speaking medical writer. The English is not too bad but there are grammatical mistakes.

A- We praise the reviewer for assessing the grammatical mistakes. A native English writer revised the manuscript.

R- INTRODUCTION

I want to stress that BAHAs or remote microphones do not restore binaural hearing. It is not clear to me that authors actually understand that detail. The only intervention that can potentially do that is a cochlear implantation in the affected ear.

A- We totally agree with this aspect, that was also discussed in the manuscript. See lines 375 and following. By the way, we stressed out this concept in the introduction, for a better clarification of this aspect. See lines 68 and 69.

R- It is also not completely clear to me the distinction between what the authors call the SSD group and the UAA group. Did CT scanning show that in the SSD group the middle and outer ear was totally normally formed. The manuscript doesn’t state anything about this.

A- we referred as SSD at a condition characterized by a severe or severe-to-profound sensorineural hearing loss on one side, with normal hearing from the other side. None of the children in SSD subgroup presented anatomic issues with their external or middle ear.  

The UAA, by the way, is the classic monoaural atresia, characterized by the typical malformed anatomy of the external and middle ear on one side only, with a pure conductive hearing loss.

We clarified this aspect editing the introduction. See lines 57 to 59. 

R- METHODS

You don’t need to state the exclusion criteria. They are literally the opposite from your inclusion criteria.

A- According to the reviewer suggestion, redundant exclusion criteria have been deleted, except for the presence of multihandicap or cognitive impairment. 

R- When you use SRT for the first time you have to say there what it stands for, not a couple of sentences later….

A- We thank the reviewer for assessing the issue. We provided a correction in the manuscript. See lines 127-128.

R- In this context you don’t talk about dB. It doesn’t mean anything. Was it dBSPL? Or maybe dBHL if your audiometer is calibrated in that way…

A- Our audiometer is calibrated in dB HL. We thank the reviewer for assessing this error. We provide a full edit of the manuscript introducing the correct unit of measure. 

R- Also make sure that when you talk about BAHA or remote microphone you talk about a hearing DEVICE. Talking about a hearing aid makes it very confusing.

A- Done. We hope that the editing of text could make it clearer. 

RESULTS

R- The analysis is only done with 10 individuals, 5 in each group. This is just too small to allow for any parametric analysis, no matter if you checked for normal distribution or not…  

A- We thank the reviewer for having found this statistical flaw. According to reviewer concerning we run a non-parametric analysis of the data.

As the reviewer can see from the edited results, not much differences in data significance have been assessed.

We edited the text and the figures according to to the non-parametric studies results. 

-------

We thank the reviewer again for its work . We hope that the paper edited according to the reviewer suggestion make that suitable for publication. 

Round 2

Reviewer 1 Report

Accept in present form.